# Deep and Frequent Phenotyping study protocol: an observational study in prodromal Alzheimer's disease

Ivan Koychev,[1] Jennifer Lawson,[1] Tharani Chessell,[2] Clare Mackay,[1] Roger Gunn,[3,4] Barbara Sahakian,[5] James B Rowe,[6] Alan J Thomas,[7] Lynn Rochester,[7] Dennis Chan,[8,9] Brian Tom,[10] Paresh Malhotra,[11] Clive Ballard,[12] Iain Chessell,[2] Craig W Ritchie,[13] Vanessa Raymont,[1,13] Iracema Leroi,[14] Imre Lengyel,[15] Matt Murray,[16] David L Thomas,[17] John Gallacher,[1] Simon Lovestone[1]

For numbered affiliations see end of article.

**Correspondence to**
Professor Simon Lovestone;
simon.lovestone@psych.ox.ac.uk

## ABSTRACT

**Introduction** Recent failures of potential novel therapeutics for Alzheimer's disease (AD) have prompted a drive towards clinical studies in prodromal or preclinical states. However, carrying out clinical trials in early disease stages is extremely challenging—a key reason being the unfeasibility of using classical outcome measures of dementia trials (eg, conversion to dementia) and the lack of validated surrogate measures so early in the disease process. The Deep and Frequent Phenotyping (DFP) study aims to resolve this issue by identifying a set of markers acting as indicators of disease progression in the prodromal phase of disease that could be used as indicative outcome measures in proof-of-concept trials.

**Methods and analysis** The DFP study is a repeated measures observational study where participants will be recruited through existing parent cohorts, research interested lists/databases, advertisements and memory clinics. Repeated measures of both established (cognition, positron emission tomography (PET) imaging or cerebrospinal fluid (CSF) markers of pathology, structural MRI markers of neurodegeneration) and experimental modalities (functional MRI, magnetoencephalography and/or electroencephalography, gait measurement, ophthalmological and continuous smartphone-based cognitive and other assessments together with experimental CSF, blood, tear and saliva biomarkers) will be performed. We will be recruiting male and female participants aged >60 years with prodromal AD, defined as absence of dementia but with evidence of cognitive impairment together with AD pathology as assessed using PET imaging or CSF biomarkers. Control participants without evidence of AD pathology will be included at a 1:4 ratio.

**Ethics and dissemination** The study gained favourable ethical opinion from the South Central—Oxford B NHS Research Ethics Committee (REC reference 17/SC/0315; approved on 18 August 2017; amendment 13 February 2018). Data will be shared with the scientific community no more than 1 year following completion of study and data assembly.

## Strengths and limitations of this study

► Participants will be phenotyped repeatedly using established biomarkers of Alzheimer's disease (AD) pathology (cognition; positron emission tomography and cerebrospinal fluid markers of pathology; structural MRI markers of neurodegeneration) as well as a number of experimental biomarkers.

► A comprehensive pilot study demonstrated that the proposed depth and frequency of phenotyping is feasible across six clinical sites and is acceptable to volunteers despite including invasive procedures such as lumbar puncture and being very demanding on participant time.

► The study data will be made available to the scientific community within one year of the study's completion.

► The engagement and involvement of stakeholders including the Alzheimer's Society and members of the public proved essential in conducting a successful pilot and will considerably enhance the conduct and outcome of the main study.

► The study size has been determined from power calculations and modelling but remains small.

## INTRODUCTION

Prevention of Alzheimer's disease (AD) is the single most important unmet need in medicine today; a common disorder with increasing prevalence in an ageing society that has huge personal and financial impact on individuals, families and societies[1] and yet without any disease-modifying therapy. The estimated global prevalence of 34 million patients for dementia as a whole in 2010, is expected to double every 20 years (to approximately 66 million by 2030 and to 115 million by 2050) due to an ageing population worldwide and improved diagnosis.[2] Trials of potential therapeutics conducted in people with established disease have failed,[3] prompting a drive

towards clinical studies in prodromal (mild symptoms not amounting to dementia) or preclinical (evidence of disease in absence of symptoms) states where intervention might be expected to be more effective.[4] However, clinical trials in such early disease states, with minimal or no symptoms, face two critical challenges; how to identify people with disease processes before symptoms become apparent and how to measure outcomes of interventions. The first of these—selection of people with preclinical or prodromal disease—has been made possible by the availability of pathology markers such as positron emission tomography (PET) imaging or cerebrospinal fluid (CSF) biochemistry, although there remain logistical and other challenges before such markers become routine outside of the context of clinical research. Second, and perhaps more challenging, is the difficulty of assessing the impact of therapeutics in clinically silent or minimally symptomatic trial participants. While study designs using progression to dementia as an outcome are appropriate for efficacy trials, such approaches are necessarily long and large and therefore extraordinarily demanding and correspondingly expensive; prohibitively so for proof-of-concept phase trials. The absence of indicative outcome measures suitable for relatively short proof-of-concept trials is therefore an undoubted limitation in the field, and acts as an impediment on early phase drug development.

The Deep and Frequent Phenotyping (DFP) study is designed to address the challenges of prodromal AD intervention development by generating a biomarker set for proof-of-concept studies early in the disease process. Multiple biomarker studies are currently underway including the Alzheimer's Disease NeuroImaging Initiative (ADNI) in the USA,[5] Australian Imaging, Biomarker & Lifestyle Study of Ageing,[6] AddNeuroMed in Europe[7] and ADNI-like cohorts are being conducted globally. In Europe, many such studies are being aggregated for combined analysis in the IMI-European Medical Information Framework (www.emif.eu). However, almost all such biomarker studies use relatively infrequent assessments and the assessments employed focus on well-established markers, such as structural imaging and molecular markers of known pathological processes.

The DFP study will combine both established and novel markers. Specifically, the study will implement PET imaging and CSF biochemistry for beta-amyloid (Aβ) plaques and tau neurofibrillary tangles, functional and structural MRI, electrophysiology for synaptic function including electroencephalograhy (EEG) and magneto-encephalography (MEG), measures of retinal pathology and a collection of biosamples for molecular biomarkers and to establish induced pluripotent stem cells (iPSCs) for in vitro studies. The study will therefore create the single largest cohort in prodromal AD individuals who have been characterised using such a wide variety of complex biomarker measures. It will also generate a comprehensive biological sample set, that when linked to the biomarker measures, will provide a platform for future translational neuroscience studies. Finally, the study will aim to take advantage of the rapidly developing information technology field to develop methods for monitoring cognition using smartphone and wearable technology. We will deploy smartphone-based cognitive applications to provide frequent testing aimed at identifying individual-specific memory extinction curves. The study will also test the utility of wearable devices to capture gait, navigational ability (indoor and outdoor) as proxies for cognitive ability and functional state. These methods have the potential to provide the high granularity, ecologically valid and intraindividual data that are likely to be required to detect subtle cognitive and functional change at scale in the trial-ready cohorts of the future.

The DFP study builds on the infrastructure of Dementias Platform UK (DPUK; https://www.dementiasplatform.uk/): recruitment will be facilitated through DPUK's information workstream for recontacting participants from its constituent cohorts and pre-existing DPUK imaging and biobank data on DFP participants will be used wherever possible. Recruitment will take advantage of the investment made in the UK in NIHR Biomedical Research Centres that are part of the Translational Research Collaboration for Alzheimer's Disease. Finally, the study will take the lead from ADNI and its partner studies, making trial data readily available to the scientific community.

## Deep and Frequent Phenotyping: a pilot study

Recognising the challenges of a study of such high intensity, we undertook a pilot DFP study to assess its feasibility and acceptability. From a feasibility perspective, the study worked to establish the operational practicability of a very extensive and repeated phenotyping over 6 months across six centres. Predetermined criteria for feasibility success included the implementation of all 10 phenotype measures with >80% completion rate in each and that data from any given modality was of at least equivalent quality to that when implemented alone in a single site (defined by variance). The pilot study recruited its first participant in December 2014 and the last study assessment took place in January 2016. The study screened 32 participants with the aim of recruiting 24 (4 per clinical site). Eight participants were excluded at screening: five as they failed to meet study entry criteria, one because of having had a recent PET scan, one for withdrawal of consent and one for an unspecified reason. Twenty-two participants were included after screening (due to two sites being unable to recruit the target number of participants in the study window) and 20 remained at 6-month follow-up (one participant withdrew consent after week 12 and one was excluded after baseline visit due to psychosis not reported at screening). The study protocol included all the assessments proposed in the current study. Completion rates for the different assessments ranged from 76% to 100% as follows: clinical assessments (100%), conventional (ie, pen and paper) and computerised cognitive assessment (100%), gait assessment (94%), ophthalmological assessment (86%), CSF collection (76%), blood collection

(95%), MRI neuroimaging (89%), neurophysiology assessment (81%). Working protocols for each modality were produced by modality leads to ensure consistency across centres and where data were successfully acquired it was of a high standard. Specifically, quality control for each modality demonstrated:

1. Clinical assessments: clinical assessments were performed at the baseline visit at all sites. Minimal errors in test completion were identified during monitoring visits and were successfully rectified.

2. Cognitive assessments: all sites performed cognitive testing with conventional assessments at baseline and day 169 while computerised assessments took place at screening and the four assessment visits. Training was required for use of the Cambridge Neuropsychological Test Automated Battery (CANTAB) test and this was provided by the manufacturer (Cambridge Cognition). Minimal completion errors in conventional testing were identified and successfully rectified. Data loss from the computerised battery was 3.6%.

3. Ophthalmology: participants had retinal imaging at three locations on two occasions: before day 85 and day 169. No training or scheduling issues were identified. Data loss was 14% due to a participant being unable to complete the task and further three participants having partial data collections due to difficulty fixating their gaze. Quality control did not reveal significant issues.

4. Gait: gait was assessed using body-worn tri-axial accelerometers at all sites in laboratory settings at baseline and day 169 as well as in free-living conditions for 7 days from day 85. Research assistants required training in the use of accelerometers and this was provided by the lead site for gait (Newcastle). Data loss was minimal (3.2%) and the received data was of high quality.

5. Blood and urine collection: blood and urine were collected at all sites using standard protocols (available on request) while peripheral blood mononuclear cell processing protocols required nursing staff training that was arranged locally. Insufficient quantity of blood for the purposes of the study was obtained in only 1% of samples. iPSC lines were successfully generated from 18 individuals.

6. CSF collection: lumbar punctures (LPs) were performed at study baseline and day 169 at all sites. Scheduling issues led to 4 LPs not taking place and a further 4 LPs were cancelled for medical reasons (two instances where LP was medically unfeasible and two where the LP at day 169 was declined due to severe headache following baseline LP).

7. MRI scanning: the design included four scans (two on sequential days during the baseline assessment episode, one at day 29 and one at day 85) at all sites. No training issues were identified. Seven per cent of scans did not take place due to scanner availability. The acquired images were reported to be of good quality with no significant intersite variability.

8. Neurophysiology: participants underwent EEG and MEG scans at four sites during the baseline assessment episode. No training requirements were identified. Twelve per cent of the scans did not take place due to scanner availability and 5% of the data was incomplete (participants unable to tolerate full length of the scans). Data quality was found to be of good quality with no significant intersite variability.

PET neuroimaging is a critical component of the proposed study and its feasibility was piloted on a single occasion before day 85 of the study at one study site only (Invicro, formerly known as Imanova). This excluded all four participants from the Newcastle site for logistical reasons. Of the 17 participants eligible for PET amyloid and tau neuroimaging, 16 completed amyloid scanning and 15 underwent tau scanning. One amyloid imaging scan was cancelled due to logistical reasons while two tau imaging scans did not take place (one participant declined to have the procedure and another scan was cancelled for logistical reasons). Amyloid PET data from one further participant was unusable due to excessive motion (88% complete data) and tau PET data from three participants were not used due to insufficient time spent in scanner (71% usable data).

The total completion data per modality is presented in figure 1. Based on a completion rate of >80% across modality and excellent quality control, the study met its criteria for feasibility success.

The acceptability of the study to participants was assessed using questionnaires, a focus group and follow-up interviews conducted by the Alzheimer's Society. Questionnaire data were collected at study end from 91% of all participants. The questionnaire asked participants to indicate the level of pleasantness or comfort that they experienced for each of the measures and provided opportunities for participants and their study partners to provide suggestions for improvements in the full study. The questionnaire also directly asked participants if they would be willing to enter a similar study and all questionnaire completers responded positively. The majority of participants expressed a sense of pride that they were doing something to benefit others. The rest of the questionnaire responses indicated a high degree of acceptability on all measures, reporting their overall experience in the study as 'good' or 'excellent', including the LP procedure.

A poststudy focus group took place at Alzheimer's Society with two members of the DFP team, Alzheimer's Society and three participants from different sites (one participant attended with their study partner). Six follow-up interviews were also carried out by Alzheimer's Society on the telephone, two of which were attended by a study partner. These activities provided qualitative confirmation of the acceptability and gave specific recommendations for the main study that were implemented in its design (more flexibility for the role of study partners, shortening the length of the PET scan and ensuring appointments are booked well in advance wherever

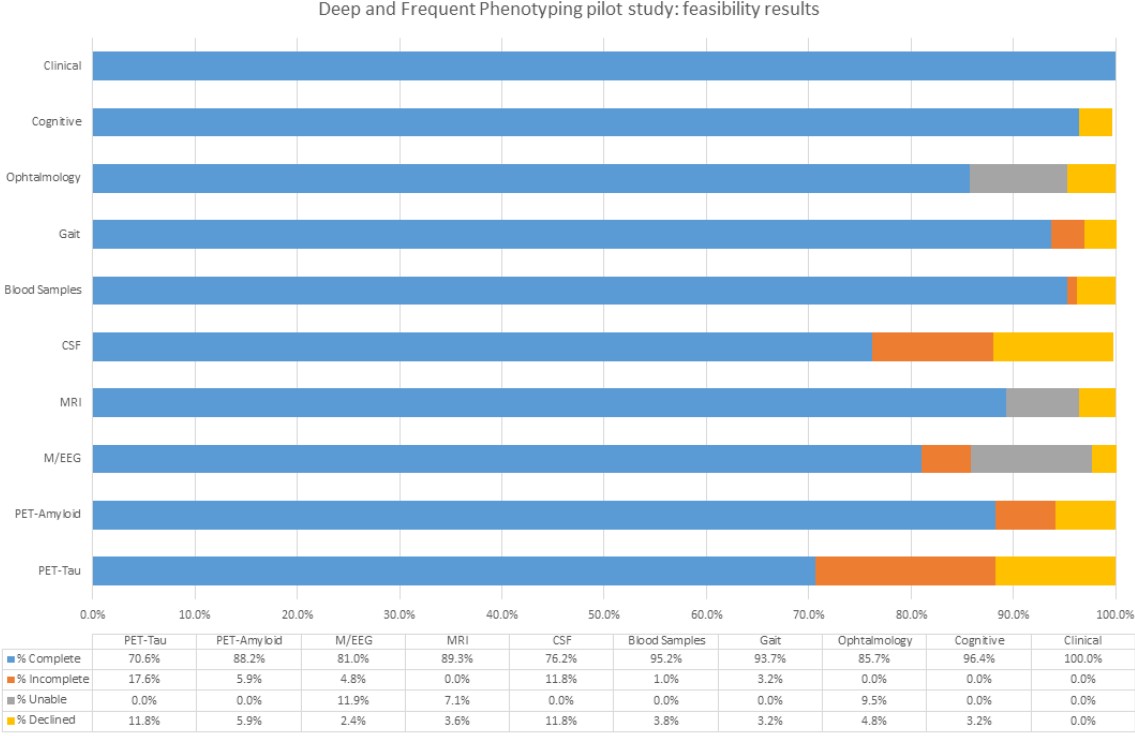

Deep and Frequent Phenotyping pilot study: feasibility results

| | PET-Tau | PET-Amyloid | M/EEG | MRI | CSF | Blood Samples | Gait | Ophtalmology | Cognitive | Clinical |
|---|---|---|---|---|---|---|---|---|---|---|
| ■ % Complete | 70.6% | 88.2% | 81.0% | 89.3% | 76.2% | 95.2% | 93.7% | 85.7% | 96.4% | 100.0% |
| ■ % Incomplete | 17.6% | 5.9% | 4.8% | 0.0% | 11.8% | 1.0% | 3.2% | 0.0% | 0.0% | 0.0% |
| ■ % Unable | 0.0% | 0.0% | 11.9% | 7.1% | 0.0% | 0.0% | 0.0% | 9.5% | 0.0% | 0.0% |
| ■ % Declined | 11.8% | 5.9% | 2.4% | 3.6% | 11.8% | 3.8% | 3.2% | 4.8% | 3.2% | 0.0% |

**Figure 1** Summary of completed assessments per modality. CSF, cerebrospinal fluid; EEG, electroencephalograhy; MEG, magnetoencephalography; PET, positron emission tomography.

possible). The qualitative research undertaken also reinforced the acceptability of LPs, with all but one participant saying it was much more comfortable than feared.

These acceptability results met predefined success criteria defined as >80% of participants willing to enter a similar study, >50% expressing willingness to proceed to a full study and there being no more than four drop-outs.

This pilot study highlighted several issues ranging from the optimal acquisition ordering of imaging protocols to avoid scan artefacts, to the optimal time of day for the various measures taking the participants' comfort into account. Working with this information and in collaboration with our user group, it was possible to improve processes, shorten timings of assessment and reduce patient burden, building in protocol flexibility where necessary. In summary, this pilot study demonstrated that a complex protocol including both complex assessments and measures perceived as invasive can be effectively established across multiple sites. Furthermore, the trial participants were sufficiently committed to remain engaged throughout the duration of the study, despite the demands of the study. In the full study, we will continue to partner with the Alzheimer's Society who will lead on the generation of participant information materials, including explanatory videos, and we will continue to work with the Alzheimer's Society lay panel to monitor study progress. We will in addition incorporate an ethical, social and legal implications workstream using qualitative research methodology to understand participant engagement and burden as well as explore the participants'

understanding and tolerability of processes related to biomarkers and precision medicine.

## METHODS AND ANALYSIS
### Study design
This is a multicentre (eight recruitment centres across the UK), non-interventional, repeated measures, observational design study in approximately 250 male and female participants aged ≥60 years with prodromal AD, defined as absence of dementia but with evidence of cognitive impairment together with AD pathology assessed using PET and/or CSF methods as well as control participants lacking cognitive impairment and without evidence of AD pathology, in a 4:1 ratio.

Repeated measures of both outcome comparator (cognition and pathology) and assessment modalities will be taken for most modalities at baseline and follow-up and at defined time points during the study and, for some measures continuously, over a period of 12 months. Modalities to be measured include: PET imaging and CSF biochemistry for amyloid and tau; functional and structural MRI; electrophysiology for synaptic function, including EEG and MEG; measures of gait and use of remote monitoring for assessment of a range of phenotypes; measures of retinal pathology and collection of biosamples.

The selection of these modalities is influenced by the decision to recruit participants with prodromal AD; the need to represent the full range of behavioural, functional

and cognitive impairments arising from AD; the need to use established assessments used in historical and current clinical trials, as well as exploratory assessments and the operational practicability of these modalities established in the pilot DFP study.

The rationale for the frequency of assessments for each modality is discussed in more detail in each modality section below and is titrated to the evidence and likely value of repeated measures. For example, very frequent repeated measures for MRI is proposed first as there is evidence of value in 24 hours repeats to enhance signal to noise and second as repeated measures at 3 months have shown change in AD, yet the optimal frequency of imaging to measure change has not yet been demonstrated. For CSF sampling, only three assessment points are proposed, as there is less evidence for change within a year.

A schedule of assessments is given in table 1.

Participants with an absence of AD pathology assessed using PET amyloid or CSF Aβ levels, who are not selected to enter the study, will be invited to participate in a substudy involving smartphone-based cognitive testing (Mezurio smartphone application) over a 12-month period. These participants will undergo a follow-up visit at the end of the testing period (clinical assessments, body fluid sampling and CANTAB cognitive testing). We anticipate to recruit 150 participants in this substudy.

### Study objectives and outcome measures

The primary objective of the study is to identify a marker set from data from a wide range of assessment modalities that tracks disease change. Modalities include markers of structure and function derived from MRI, markers of synaptic function derived from electrophysiology (EEG and MEG) and markers from innovative methods including assessment of retinal pathology and both gait, cognition and activity from wearable devices and connected technology together with biosamples. These samples will be made available for future studies using metabolomics, genomics, proteomics, transcriptomics and other approaches. We plan to meet the primary objective by using measurements of a change in comparator modalities of pathology (PET or CSF measures) or change in cognition.

The study's secondary objective is to identify a marker set from summary data from a wide range of assessment modalities that predicts disease change. The comparator measure for this objective would be measurement of a change in cognition and transition from prodromal to disease phase at 1 year and beyond.

### Participant selection

The study aims to recruit male and female subjects aged >60 years with prodromal AD, defined as absence of dementia but with evidence of AD pathology as assessed using PET or CSF (ie, amyloid-positive individuals or AmyPos); and amyloid-negative subjects (AmyNeg) without AD pathology on PET imaging or CSF analysis, in

a 4:1 ratio. Inclusion and exclusion criteria for this study are presented in online supplementary materials.

In order to identify such individuals we will use a screening funnel approach, with multiple steps, seeking people that have agreed to be approached for research participation from research cohorts, those signed up to research databases and from memory clinics. To reduce screen failure when trying to identify AmyPos and AmyNeg individuals in a 4:1 ratio, we will use existing data including cognitive scores and family history of dementia where it exists, such as when seeking participants from cohort studies. In regard to the cognitive scores we will define subclinical impairment as being in the range between 1 and 1.5 SD below the mean for the test used in the respective cohort. Where there is no recent cognitive data available, we will ask cohorts to select on age and family history and invite those participants. In the second step we will then ask those participants that respond to complete a brief cognitive test online (CANTAB Paired Associates Learning [PAL] Task), via the study website (www.dfpstudy.co.uk). Enrolment will include people with subclinical cognitive impairment (defined as scores ranging between 1 and 1.5 SD below the mean for the age-adjusted and sex-adjusted mean of the cognitive test) and no apparent impairment (defined as scores in the range between 0.5 and 1 SD below the mean) in a 4:1 ratio, in part to avoid disclosure of cognitive state. Where there is no suitable prior information, such as in some research-willing registers or in direct approaches to the public through social media and advertisement, we will invite people within the age range of the study and with a family history of dementia to complete the same online-based prescreening methods. Potential participants attending memory clinics with a diagnosis of mild cognitive impairment, who express interest in participating in research, may also be approached if they meet criteria.

In the final step of the screening funnel, people identified through the process described above will be invited to attend a screening appointment where inclusion and exclusion criteria will be reviewed and apolipoprotein E (APOE) status will be established (either through consent to access parent cohort data on APOE or providing sample for APOE genotyping). An APOE-4-enriched cohort (ie, 4:1 ratio of APOE carriers vs APOE non-carriers) will be invited to a second screening appointment where cerebral amyloid status will be determined using PET imaging or CSF sampling. Final inclusion in the study will be based on amyloid positivity at a ratio of 4:1. A summary of participant selection is provided in figure 2.

### Study procedures

Participants will be interviewed with a semi-structured clinical assessment for sociodemographic and lifestyle factors, and family history of AD/dementia in first-degree relatives. We will record medical history and medication, together with measurements of height, weight, waist-to-hip ratio and blood pressure. Assessment of cognition and function will include Mini Mental State Examination,

**Table 1** Schedule of assessments

| | Screening | Assessment phase | | | | | | | |
|---|---|---|---|---|---|---|---|---|---|
| **Visit** | 1* | 2† | 3† | 4 | 5 | 6 | 7 | 8 | 9 |
| **Visit type** | Screening | Baseline | Baseline | 1 Month | 2 Months | 6 Months | 8 Months | 10 Months | 12 Months |
| Time (months) | −2 | 0 | 0 | 1 | 2 | 6 | 8 | 10 | 12 |
| Time (days) | −60 to −2 | 1 | 2–5 | 30 | 60 | 180 | 240 | 300 | 365 |
| Visit windows (days) | – | – | +3 | ±14 | ±14 | ±14 | ±14 | ±14 | ±14 |
| **Procedures** | | | | | | | | | |
| Consent | X | | | | | | | | |
| **Clinical and cognitive measures** | | | | | | | | | |
| Clinical interview | X | X | | | | | | | |
| Clinical assessments | X | X | | | | | | | X |
| CANTAB cognition battery | X | X | | X | X | | X | X | X |
| EPAD cognition battery | | | X | | | X | | | X |
| **Specimens** | | | | | | | | | |
| CSF sampling | X‡ | | X‡ | | | X | | | X |
| Blood, saliva and urine | X | X | X | X | X | X | X | X | X |
| Visit | 1* | 2† | 3† | 4 | 5 | 6 | 7 | 8 | 9 |
| **Imaging** | | | | | | | | | |
| PET amyloid | X§ | | | | | | | | X§ |
| PET tau | | | X¶ | | | | | | X§ |
| MRI | X** | X | X | X | X | X | X | X | X |
| MEG and EEG | X | X | | | | | X†† | | X |
| **Wearable technology and connected devices** | | | | | | | | | |
| Gait and peripherals‡‡ | | X | | | | X | | | X |
| Mezurio smartphone application§§ | | X | X | X | X | X | X | X | X |
| Indoor localisation system | | X | X | X | X | X | | | |

**Table 1** Continued

| Visit | Screening | Assessment phase | | | | | | | | |
|---|---|---|---|---|---|---|---|---|---|---|
| | 1* | 2† | 3† | 4 | 5 | 6 | 7 | 8 | 9 | |
| Visit type | Screening | Baseline | Baseline | 1 Month | 2 Months | 6 Months | 8 Months | 10 Months | 12 Months | |
| Outdoor localisation system | | X | X | X | X | X | X | X | X | |
| Mood and sleep ratings | | | X | X | X | X | X | X | X | |
| **Other** | | | | | | | | | | |
| Ophthalmological assessment | | X | | | | X | | | X | |
| Participant acceptability | | | | | | | | X | | |

*Screening visit may take place over several appointments.

†Baseline assessments can be performed on Day 1 or Days 2-5 with the exception of PET tau, which can be performed on Days 2-30. See section 11 for further details.

‡CSF Sampling may be carried out as an alternative to PET Amyloid at Screening. In those completing CSF sampling at screening, no additional CSF sampling will be performed at baseline.

§For participants screened using CSF a single PET Amyloid scan will be done at any point during the 12 month study duration.

¶A subset of participants (n=100) will undergo follow-up PET Amyloid at approx. 12 months with PET Tau at baseline and again approx. 12 months later.

**It may be possible in some centres to include one MRI scan at screening, to reduce assessment burden at baseline.

††The MEG and EEG scan at 8 months will be undertaken on participants in some study centres only, subject to funding and capacity. It will be made clear to a participant whether 2 or 3 M/EEG scans are proposed for them.

‡‡7-day gait and continuous assessment with devices for cognition.

§§Mezurio interactions will not coincide with the 7-day gait assessments.

CANTAB, Cambridge Neuropsychological Test Automated Battery; CSF, cerebrospinal fluid; EEG, electroencephalograhy; EPAD, European Prevention of Alzheimer's Disease; MEG, magnetoencephalography; PET, positron emission tomography.

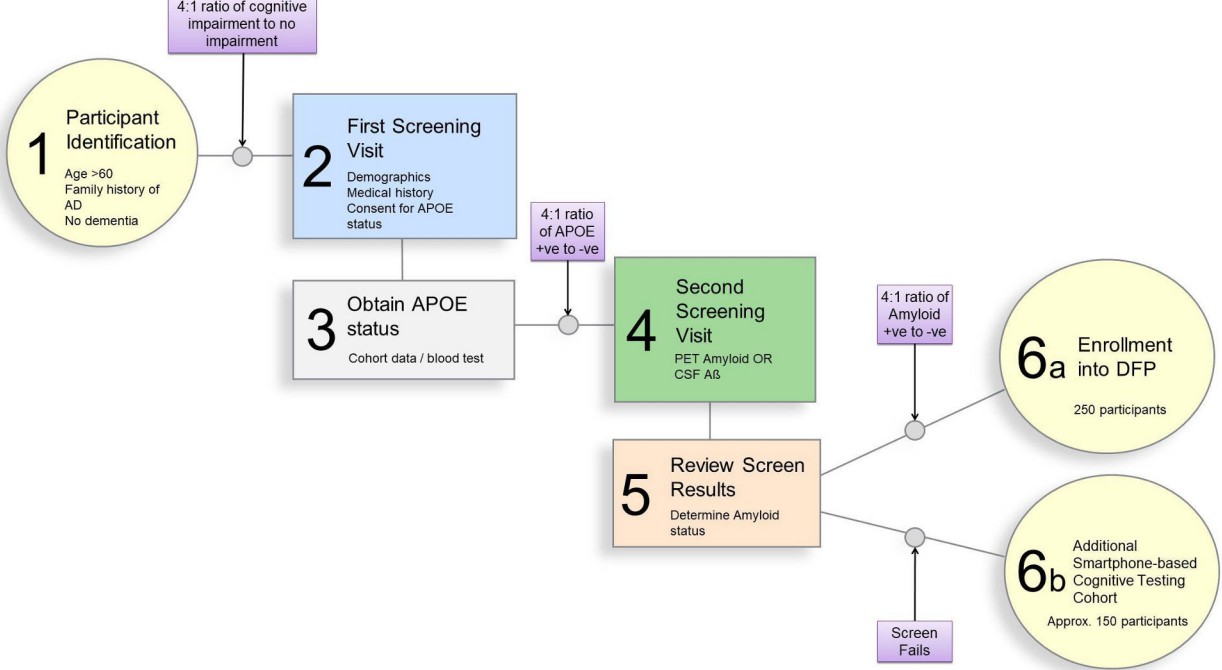

**Figure 2** Summary of participant selection. AD, Alzheimer's disease; APOE, apolipoprotein E; CSF, cerebrospinal fluid; DFP, Deep and Frequent Phenotyping.

Amsterdam Instrumental Activities of Daily Living questionnaire, The Brain Injury Screening Questionnaire, the National Adult Reading Test, the Mild Behavioural Impairment Scale, together with the Clinical Dementia Rating. The assessments will also include Geriatric Depression Scale, State-Trait Anxiety Inventory and Pittsburgh Sleep Quality Index.

## Cognition

For the purposes of website-based cognitive screening (where pre-existing cognitive scores are not available), we will use a browser-optimised version of the PAL task that is part of CANTAB (Cambridge Cognition). PAL total error scores will be used to define cognitive impairment (scores between 1 and 1.5 SD below the age-adjusted and sex-adjusted mean) and no apparent cognitive impairment (scores between 0.5 and 1 SD below the age-adjusted and sex-adjusted mean) for the purposes of inviting participants to the first screening appointment. The PAL scores will be deleted once this information has been used to determine which participants will be invited to the screening appointments.

In the main study, cognition will be assessed using two batteries of cognitive tests. At baseline, 6 months and at 1 year, the protocol for assessment of cognition established for the IMI-European Prevention of Alzheimer's Disease (EPAD) study will be implemented. This protocol was established by an expert working group[8] and consists of:

▶ Reaction Time/Information Processing Speed/ Conceptual Shifting/Selective Attention including

the Eriksen Flanker Task (NIH EXAMINER/Toolbox) and Coding (RBANS);
▶ Verbal Episodic Memory including List learning (RBANS) and Story Memory (RBANS);
▶ Visuospatial Analysis including Figure Copy (RBANS) and Semantic Fluency (RBANS);
▶ Working Memory including Digit Span (RBANS) and Dot Counting (NIH Examiner);
▶ Allocentric Spatial Memory: Four Mountains Task (UCL/University of Cambridge);
▶ Paired-Associate Learning: Name-Face Pairs (UCSF);
▶ Navigation in Egocentric Space: Virtual Reality Supermarket Trolley (UCL).

In addition, a tablet-based cognitive test battery using the CANTAB (Cambridge Cognition)[9–11] will also be implemented at the following time points: screening, baseline, day 30, day 60, day 240 and day 300. This battery focusses on five neuropsychological domains: attention, episodic memory, processing speed, working memory and executive function, identified by the National Institute of Health Cognitive Toolbox as being the core determinants of functioning and independence.

The CANTAB battery consists of:
▶ Rapid visual processing task (attention and processing speed);
▶ PAL task (episodic memory);
▶ Reaction time task (processing speed);
▶ Spatial working memory task (working memory and executive function).

The tablet-based CANTAB battery was successfully used in the pilot DFP study at five time points. A short

questionnaire after each testing session ascertained ease of use from the subjects' perspective. Interim analysis of the data showed that the cognitive tests produced results consistent with expected levels of performance in this population and the data received were comparable across sites. All of the tests had a completion rate of between 95% and 100%, with the battery overall having a completion rate of 97%.

## PET imaging

PET Aβ tracers have been established over the last decade and Amyvid, Neuraceq and Vizamyl have been approved by the US Food and Drug Administration and European Medicines Agency. Radiotracers made available in recent years (such as [18F]AV-1451, [18F]MK-6240 and [18F]PI-2620) enable in vivo imaging of tau pathology deposits.[12][13] Initial data from these tracers has demonstrated a tau signal, which increases when moving from healthy controls, through MCI and into AD.[14][15] Understanding the relationship between Aβ and tau during the earliest phase of the disease is likely to be critical for the successful development of therapies.

PET Aβ will be the primary method for screening participants. Where for logistical reasons CSF Aβ was used instead of PET Aβ for screening, a single PET Aβ scanning will be completed at any point during the study. A subset of 100 subjects will have both Aβ and tau scans at baseline and at 12 months follow-up. For each PET scan, participants will receive an intravenous injection of the appropriate radiotracer with data acquisition in the region of 20–30 min (depending on tracers) following an uptake period appropriate for the particular radioligand and its kinetics. All PET scans will be performed as outpatient procedures on either a PET/CT or PET/MR scanner. In terms of radiation exposure participants will undertake a maximum of four PET scans over the course of the study, with a maximal injected dose of 150 MBq per scan. The exposure varies between participants depending on which PET tracers they receive, whether they undergo Aβ screening or Aβ and tau tracer PET imaging with follow-up PET scans, and to a lesser extent whether they undergo PET/MR or PET/CT imaging. The total effective dose therefore varies between 2.8 mSv (for a single Amyvid PET-MR scan) and 15.0 mSv (for Neuraceq and [18F]AV-1451 PET-MR scans and potential attenuation correction CT scans at baseline and follow-up). These doses are equivalent to about 1.5 and 6.5 years of average background radiation in the UK respectively and have been chosen to provide acceptable image quality while minimising the total radiation exposure.

Data will be analysed centrally, as in the pilot study, including image-based motion correction, region of interest definition and generation of regional standardised update ratio values (SUVR). Parametric SUVR images in stereotaxic space for voxel-wise analyses will also be generated. Aβ positivity will be determined automatically based on the whole cortex SUVR value according to literature-based threshold for that tracer. Higher-level study analyses will then be performed in image space to explore the relationship between Aβ and tau, the change in protein densities over time and their relationship to other clinical assessments and biomarkers.

## Magnetic resonance imaging

Advances in image acquisition and analysis of MRI now facilitate a diagnosis of AD at a predemensia (eg, mild cognitive impairment) stage and have been used to monitor change in interventional studies. This is recognised in the incorporation of imaging into international diagnostic guidelines for AD and prodromal AD.[16] In particular, measures of whole brain and hippocampal atrophy have been shown to precede symptoms by several years and longitudinal imaging provides power to detect change in these early cases.[17]

MRI will be performed on five occasions: day 1, days 2–5, day 60, day 180 and day 365. In some centres, it may be possible to combine baseline MRI with screening PET, reducing the overall scan time. The MR acquisition protocol will consist of the EPAD core sequences (ADNI 3-matched volumetric T1-weighted imaging, 3D T2-FLAIR, axial T2-weighted and T2*-weighted imaging), and a set of more advanced sequences: 3-shell HARDI diffusion-weighted imaging with b=300, 1000, 2000 s/mm$^2$, resting state functional MRI, quantitative susceptibility mapping (QSM) and arterial spin labelling (ASL) to provide measures of macrostructural and microstructural loss and functional network disruption.

Analysis of the serial MR imaging protocol will be similar to that used in the pilot DFP study, comprising measures of:
► Macrostructural change derived from volumetric T1-weighted imaging with within-subject registration and quantification including global atrophy rates (whole brain and ventricular volume), regional (hippocampal) atrophy rates and cortical thickness and template-based regional (parcellation) atrophy and data-driven model-dependent analyses of shape change across multiple grey matter regions.
► Microstructural pathology and loss of connectivity—change in grey and white matter diffusivity (DWI-derived fractional anisotropy, axial and radial diffusivity).
► Connectivity and change in network status (from resting-state fMRI).
► Vascular burden and microhaemorrhages—secondary to both atherosclerotic and amyloid angiopathy (eg, derived from FLAIR, T2-weighted and QSM).
► Perfusion and arterial arrival times from ASL.

The total scan time per visit will be approximately 1 hour.

To model physiological noise and instrument variability, baseline data will be acquired twice as previously established as a noise reduction paradigm,[18] followed by closely spaced assessments (day 60 and day 180) to establish the lower limit of MRI temporal sensitivity and at day 365. Assessment of the day 365 scan will benefit from multi-level modelling of within-subject and between-subject

variance for each of the biomarkers, such that rates of change (eg, of atrophy) will be specified more precisely. In combination with parallel initiatives (eg, ADNI 3), this will allow sample size estimates for clinical trials over a range of intervals to be determined.

## Neurophysiology

Changes in Aβ and tau lead to remodelling and loss of synapses and microcircuits, with impaired synaptic function and plasticity-dependent learning and memory.[19] It is widely hypothesised that synaptic loss and dysfunction precedes the neuronal death that eventually results in gross structural changes seen in MRI. The study will determine whether the impact of peri-symptomatic AD pathology on synapse function and circuits is identifiable using MEG-EEG.

All subjects will undergo two MEG/EEG sessions (baseline and 1 year) and a subset (based on centre capacity) may undergo a third session at 8 months. At each session, the experimental measures will be standardised, and include:

► Acute plasticity supporting predictive coding of sensory regularities, probed by an auditory mismatch negativity task;
► Evoked and induced MEG/EEG signatures of multimodal associative learning;
► Evoked and induced MEG/EEG signatures of scene repetition learning;
► Evoked and induced MEG/EEG signatures of an audio-visual-motor task;
► Visual evoked gamma in posterior cortex;
► Resting-state indices of large-scale network dynamics, including (a) partial directed coherence of MEG/EEG and (b) fast transient dynamics and variability in large-scale brain networks.

The MEG/EEG scans will be performed in the same order, with up to 60 min of recording in a series of short 'tasks' each lasting 5–20 min, including:

► Simple audio-visuo-motor task: participants press a button quickly in response to a simple black and white visual stimulus presented on screen together with an auditory tone.
► Scene repetition task: a series of landscape scenes are viewed on screen, some of which are repeated after a variable interval. The participant is asked to look out for an occasional prespecified target (an image of the moon), and press a button when this is seen.
► Cross modal learning: the participant views a series of objects on screen, which are paired with an associated tone. The learnt associations between object and tone are revealed by low frequency presentations of novel tones and of mismatching object-tone pairs.
► Resting eyes open: participants will be instructed to relax and look gently at a fixation cross.
► Auditory mismatch-negativity task: a brief auditory threshold paradigm is run and the main task tones are adjusted to be 60 dB above the estimated auditory threshold. The auditory mismatch presents variable short 'beep' tones every 500 ms. No response is required of the participant, who may view a natural history film without sound to maintain wakefulness and reduce tiredness.

## Wearable technology and connected devices

Wearable technology and connected devices are defined as devices worn on the body or embedded into mobile phones, watches, bracelets, etc. While such technology has become pervasive and is increasingly used by the general public, there are few examples of its effective use in dementia research to date. There is much promise that such devices could be sensitive to subtle changes in cognition and provide opportunities for unobtrusive monitoring in the participants' own environment. An important goal therefore for the DFP study is to assess the ability of such devices to measure behaviours or phenotypes as potential measures of change in prodromal AD alone or in combination with established or experimental assessments.

### Quantifying gait using wearable technology

Gait is increasingly recognised as an important tool to inform diagnostic algorithms, track disease progression and measure the efficacy of interventions in a range of neurodegenerative diseases, including AD.[20] In the pilot DFP study, we deployed a body-worn tri-axial accelerometer for detailed gait analysis in the clinic and unobtrusive monitoring of gait and activity during free living.[21] Participants with AD showed a more variable gait in the clinic than during free living which by contrast revealed differences for temporal features of gait. These preliminary results indicate a subtle but important effect of environment or habitual behaviour which will be examined in full in this study.

Gait will be assessed on three occasions in the clinic and through 7-day home monitoring at baseline, day 180 and day 365. For home monitoring, the monitor will be attached during the clinical visit and worn for 7 days, after which time it will be returned. Assessment of gait in the clinic will include a validated single and dual-task gait protocol that takes approximately 30 min to administer, using a small, lightweight and waterproof accelerometer. Assessment of gait in the home will involve participants wearing the accelerometer on their lower back for 7 days. Gait outcomes will be derived from a theoretical model and will include a range of outcomes known to be sensitive to cognition.

### Continuous assessment of function and cognition using mobile technologies

Three complementary methodologies will be deployed across the DFP sites to explore and identify cognitive and/or functional change longitudinally: cognitive smartphone assessments; localisation methodology using smartwatches and brief mood and sleep assessments. All data will be anonymised. Several of these approaches require the participant to have an iOS or Android smartphone.

Where participants do not have such a device it will be provided.

i. Mezurio smartphone application

The Mezuirio smartphone application is a collection of cognitive tasks designed for relevance to AD. Integrated within a single application are tasks targeting: episodic memory, semantic memory, executive function, language, visuospatial ability and speed of processing. All tasks are scheduled within a single application to simplify the installation process and user-interface. Interactions with the tasks within Mezurio are positioned at approximately monthly intervals, with more concentrated testing sessions at baseline and in the final month of the study. The maximum anticipated duration of engagement with each task across the trial is shown in table 1.

Pseudonymised data collected from the app will automatically upload to secure study servers hosted by Oxford University Medical Sciences IT Services with anonymised summary data then being periodically uploaded via a secure manual procedure to the DPUK portal.

ii. Sea Hero Quest

Sea Hero Quest (http://www.seaheroquest.com/) is a tablet or smartphone-based virtual reality test to examine spatial navigation and orientation behaviour. The game has been played by over 2.7 million people, generating over 73 years of gameplay, making it the largest dementia study in history. The gameplay presents participants with different navigational challenges, including i) wayfinding (ie, navigating to different checkpoints based on initially presented map information); ii) path integration (ie, navigating to a waypoint from where the player launches a flare towards the starting point of navigation); iii) radial maze (ie, maze navigation according to the landmarks or internally generated information). All tasks measure different combinations and delineation of allocentric and egocentric navigation behaviour. Participants only play selected levels of the game.

We will ask participants to consider playing Sea Hero Quest as part of their involvement in this study. This will be an optional task. We will ask participants to play the game at least once during their participation in DFP, for at least 15 min. This will provide enough data for comparison against average performance in existing datasets. Data transfer from Sea Hero Quest to DFP will be processed to be pseudonymised and no personal identifiable information will be exchanged.

iii. Localisation system

We will deploy methods from the emerging field of Positioning Systems to provide a continuous view of participants' movement and activity within and outside of their homes. In terms of indoor positioning, this novel method solves the problem of insufficient indoor satellite coverage and limited use of handsets indoors and provides information on patterns of activity within the day, indoor walking speed and time spent within specific rooms. We plan to generate a pattern of activity within the home for each participant and during the 6 months of indoor positioning data recording (first 6 months of

the study or day 1 to day 178), we will seek to identify deviations from this pattern as early signs of deterioration in function. Mobile outdoor tracking will be recorded throughout the 12-month study period and will capture GPS data outside of the home, to test the hypothesis that the disruption of entorhinal cortex and hippocampal function in early AD will cause progressive disintegration of the 'cognitive map', identifiable on analysis of everyday spatial navigational behaviour.

We will deploy commercially available smartwatches for high-frequency movement data collection. Participants will be encouraged to charge the smartwatch every night, and then carry the device with them as soon as they get up. Participants will be given 4–6 Bluetooth mains powered beacons, which should be distributed within specific rooms of their homes (eg, living room, kitchen, etc). This will create fixed navigational points for the smartwatch to wirelessly identify. The smartwatch data will be automatically uploaded anonymously and with encryption when in range of known wireless internet to a secure server hosted by the Oxford Department of Computer Science. Mobile outdoor tracking will be recorded through an application installed on participant's smartphones. It will track GPS location and inertial sensor data which will tell us about their movements outside the home. The app will periodically synchronise with the secure study servers over the participants' wireless internet. Anonymised data from the indoor and outdoor tracking applications will be uploaded to the Dementias Platform UK wearable devices portal via a secure manual procedure.

From indoor positioning data we will derive a summary of indoor domestic movements (eg, proportion of time spent at various rooms, instances of room re-entry), inferences about the type of activity over a period (eg, sitting, walking, cooking, etc) and characteristics of those activities (eg, in-home walking speed). We will identify a pattern of activity within home-based daily routines on an individual participant level using a sequence matching algorithm and will seek to identify deviations from this pattern longitudinally as a marker of functional deterioration.

Outdoor navigational behaviour will be analysed at differing levels of complexity, ranging from simple measures (eg, numbers of destinations visited) through to low level topological (eg, note degree, average path length) and higher level topological (eg, clustering coefficient, local and global efficiency metrics).

### Mood and sleep diary

Using an app developed by the University of Oxford, we will collect information about participants sleep and mood at different points of the day. Brief ratings scales will be used to assess individual mood components, and sleep quality and duration. To minimise the participant setup burden, these Mood and Sleep tasks will be deployed within the Mezurio app described above. Data will be uploaded through a cloud-based transit layer (which will comply with UK Data Protection Act standards) then be held on servers hosted by the Oxford

University Medical Sciences IT Services, and managed by the Oxford Cognitive Health and Neurosciences Clinical Trials Unit. Analysts will use both tools developed in Oxford, and tools hosted outside Oxford which comply with the UK Data Protection or US-EU privacy shield. All summary measures resulting from these analyses will be uploaded to the Dementias Platform UK wearable devices using secure manual procedures. Raw diary data will be destroyed once summary measures have been finalised.

### Ophthalmological assessment

Recent evidence from dementia studies has demonstrated changes in retinal nerve fibre layer and overall retinal thickness, retinal vascular calibre in the macular area[22] as well as the appearances, and progression of extracellular deposits under the retinal pigment epithelium of the peripheral retina.[23]

In this study, we will combine basic assessments of visual acuity at the time of recruitment with structural retinal imaging and tear collection in both eyes. Changes in the central and peripheral retina will be assessed by Ultra-Wide Field Imaging (UWFI) using a laser scanning ophthalmoscopy (OPTOS). Colour and autofluorescent images will be used to assess extracellular deposit formation, cell atrophy and vascular changes both in the macula and the periphery.[23 24] In addition, optical coherence tomography (OCT) images will be generated to assess changes in nerve fibre and other retinal layers around the papilla as well as in the macula, and to measure changes in the choroidal thickness.[25] Tear collection samples will be stored at −80°C and use for proteomics[26] and metabolomics[27] at a later stage. The acquisition of UWFI and OCT images and tear collection will take 30 min in total. The ability to non-invasively detect retinal amyloid load is an exciting additional possibility.[28] We will use an approach developed by Neurovision Imaging utilising laser scanning ophthalmoscopy to detect amyloid deposits in the human retina. The imaging for amyloid index will add a further 30 min to the imaging protocol.

Ophthalmological assessments will be carried out at baseline (days 1–5), day 180 and day 365. Subjects will be dilated with tropicamide after tear collection and before imaging to enhance image quality and field of view. Therefore, the total assessment period will take approximately 60 min. Following their acquisition, anonymised images will be transferred to the CARF image analysing centre at Queen's University Belfast, and finally grading results will be incorporated into the DFP database.

### Molecular markers
#### Cerebrospinal fluid

The pilot DFP study demonstrated that repeated LP for CSF is practicable and acceptable to elderly people in the UK, especially when performed with fine gauge needle aspiration. A CSF collection and processing protocol widely used in clinical studies[29] was adapted for use in the pilot DFP study. Seventy-six per cent of LPs were successfully conducted. On two occasions subjects declined to

have their second LP because of experiencing headache at the first CSF collection. A further two could not be completed for medical reasons. Four LPs were omitted due to logistics and not participant acceptability problems.

In this study, LP will be performed at baseline, at day 180 and at day 365. We will measure Aβ and tau, using the best available assay, at a central laboratory and in addition aliquots of CSF will be stored for future exploratory studies. A maximum of 10 mL of clear CSF will be collected on each occasion in accordance with local safety procedures.

#### Blood, saliva and urine

Sample storage will be centralised at an established national resource centre following processing and aliquoting. Sample storage will meet the highest standards for sample safety and preservation, and compliance with requirements of the Human Tissue Act. Samples including blood, CSF, urine and saliva will be made available to DFP study investigators and the wider scientific community via a sample access committee for use in future research projects that have appropriate approvals in place. We anticipate applications to use this valuable sample using untargeted methodologies for exploratory studies such as proteomics, metabolomics and genomics and targeted methodologies to validate or otherwise confirm previous findings from such exploratory studies.

### Participant acceptability

Identifying expectations of participation in research is particularly pertinent in this study, as there is a significant burden on subjects, with important implications for recruitment and drop-out.

In the pilot DFP study, subject acceptability was assessed by considering recruitment, participation and drop-out due to study factors. In addition, a subject engagement study was conducted by the Alzheimer's Society; questionnaires were designed to be completed by subjects at their final study visit, a focus group and six telephone interviews were conducted with subjects and their carers. A further focus group was held at the end of the study. The questionnaire directly asked subjects if they would be willing to enter a similar study; all subjects that completed a questionnaire answered 'yes'. Subjects indicated a high degree of acceptability on all measures, reporting their overall experience as 'good' or 'excellent'. Recommendations for improvements to the study from subjects and carers will be incorporated into this study and include: more flexibility for the role of carers or study partners in the study; shortening the length of the PET scan and ensuring that appointments are booked well in advance whenever possible.

Understanding the experience of participation in this study will contribute to debate on two important issues in research. First, this study is representative of a trend in biomedical research towards the development of 'precision medicine' platforms based on the deep biological profiling of disease processes and it is imperative to

understand how participation in such initiatives affects subjects and how their experience shapes the research process. Second, such work contributes to clarifying the concept of 'participant burden' in clinical research, something that remains ill-defined and poorly understood. This component of the study builds on the pilot work highlighted above and work within the ethical, legal and social implications work package of DPUK where specific scenarios within the DFP study were explored in workshops with potential subjects.

In-depth qualitative interviews will be conducted with up to 30 participants at two time points: within 14 days of the baseline visit and within 14 days of the day 180 visit. Interview participants will be approached during periods of ethnographic observation at three of the research sites (selected to include both rural and urban populations) during baseline and day 180 visits. Observation and interview data will provide the foundation for an empirically informed assessment of participant's expectations and experiences of participation. This questionnaire will explore participation burden with all study participants at day 300.

## Sample size calculation and data analysis plan
### Sample size calculation
We performed power calculations estimating the numbers needed to identify a biomarker of change from summary measures. Assuming that (i) 20% of the APOE-enriched prodromal population would show deterioration in cognition at 1 year; (ii) 80% power to detect a difference in outcome between those with worsening cognitive functioning and those without of at least 0.7 SD (or 0.8 or 1) of 1 year change in biomarker values; (iii) there are 10 of the biomarker summary values with effect sizes of at least 0.7 (or 0.8 or 1) and (iv) a false discovery rate of 5% using a two-sided, two sample t-test, then a total sample size of 225 (175 or 115) would be required. For a single test, the individual significance level is set to approximately 0.001. The probability of detecting all 10 tests where the true standardised effect size is at least 0.7 (0.8 or 1) is 0.12 (0.13 or 0.14). This calculation is in line with observed figures from ADNI, where it can be shown that a sample size of 186 is sufficient to detect a 25% change in marker values over 1 year in early and prodromal AD.[30]

Therefore, based on power calculations above and building on the data from structural imaging we aim to include 250 participants, 80% being amyloid positive. Such a trial will have sufficient power to detect minimal change in established markers in 1 year and therefore suitable for identification of biomarkers of greater efficacy. Using the screening algorithm discussed above and described in figure 2, we estimate we will need to screen 360 individuals to achieve this goal.

We may recruit additional participants if some drop out to ensure the final number of participants is as close to 250 as possible.

### Data analysis plan
In this section, we broadly describe the statistical methods likely to be used in this study in the first wave of analysis. However, we expect many other approaches to be used as the data become available to the wider scientific community. As there will be a wealth of data collected from various different modalities (imaging to molecular markers), our data analysis strategy begins by performing separate analyses by modality using the standard suite of statistical tools and software developed for these modalities. These analyses will include generating the summary (derived) data from the raw data using well recognised, tested and/or standardised pipeline/protocol in order to extract summary features of importance (eg, regions of interest).

First wave analyses will be informed by the pilot DFP study and will begin by comparing the average 1-year change in biomarkers between those who had a worsening in cognitive functioning (or with a change in disease state) and those who did not, using only the baseline and 1-year measurements of these biomarkers. Cognitive functioning will be defined through the cognitive coprimary outcome measures of i) adjusted errors on the PAL task and ii) between search errors on the spatial working memory (SWM) task. Clinically meaningful deterioration will be defined as a loss of either 3 or 1 points (PAL and SWM, respectively) over the 12-month study duration. While normative performance for PAL and SWM has been defined as 0.6 and 0.4 points change per year of age, respectively,[31] in amyloid-positive and APOE E4 carriers an average 3-point change in PAL and 1 point in SWM has been observed for a sample of MCI patients who transition to AD.[32] Further investigations into identifying a marker set from a wide range of assessment modalities that tracks disease change will use regression methodology (including mixed models and extensions) that incorporate change measures and may involve dimension reduction techniques (eg, principal component analysis), variable selection approaches (eg, LASSO and Elastic Net) and efficient use of repeated measurements. For example, linear mixed effects analyses of the whole group on the longitudinal CANTAB composite memory score will be used to determine if there are relationships with the various markers. Similar regression methodology will be used to identify a marker set that predicts disease change. Here however, previous measurements of the markers will be considered instead of change measures. Alternative machine learning methodology for unsupervised and supervised learning, which are now regularly used for analysing Big Data problems, will also be used. Issues related to missing data, multiple testing, validation, correct functional forms of variables, high correlation of biomarkers, the potential differing timing pattern of biomarkers and cognitive outcomes will be addressed.

Additional analyses include repeatability/reproducibility of MRI, the interchangeability/calibration of CANTAB and EPAD batteries of cognitive tests using the baseline data (linked to the inclusion of DFP data from

participants who are also part of EPAD into the EPAD's disease modelling) and joint modelling of biomarkers (such as between Aβ and tau).

## Ethics and dissemination
### Subject confidentiality
The study staff will ensure that the subjects' anonymity is maintained. The subjects will be identified only by a subject ID number on all study documents and any electronic database (except for the participant recruitment website run by TrialSpark), with the exception of the electronic Clinical Report Form (eCRF), where subject initials may be added. All documents will be stored securely and only accessible by study staff and authorised personnel.

Subject data will be deidentified at all times. The study will comply with the General Data Protection Regulation, which requires data to be anonymised as soon as it is practical to do so. Measures to maintain confidentiality with wearable modality assessments are discussed in 'Ophthalmological Assessment' section.

### Data monitoring and access
Direct access will be granted to authorised representatives from the sponsor and host institution for monitoring and/or audit of the study to ensure compliance with regulations.

### Data management
Clinical data collection will be using electronic clinical record forms and this together with summary data generated from each of the modalities will be collected centrally. Trials management software together with a website bespoke to different routes of participant identification will be used.

### Open data sharing
Data sharing between the DFP project and the scientific community will be enabled on the Synapse infrastructure (Sage Bionetworks, Seattle, USA), no more than 1 year following study-team exclusive access to the fully assembled dataset (we anticipate requiring under a year to collate and prepare the data for analysis). Synapse operates under a governance process that includes well-documented Terms and Conditions of Use, guidelines and operating procedures for handling data, data security measures with strict information and privacy enhancing technologies, as well as the right of audit and external reviews (WIRB 20112068). The DFP study consent will include agreements for data sharing and will be transparent and explicit in the intent for this study to share pseudonymised data within the large multisite study team and to share fully anonymised data more widely with the scientific community.

Once data curation has completed the data will be made available to qualified investigators through a public DFP portal that is linked to the DPUK portal. Data access will require a data use statement that will be posted together with the investigator name and institution on the public DFP data release site, enabling others to see how the data are being used. The file annotations will facilitate the incorporation of data generated through the DFP with other AD-related studies hosted in synapse, such as data generated through the Accelerating Medicines Partnership-Alzheimer's Disease (synapse.org/ampad) and the AddNeuroMed study (https://www.synapse.org/#!Synapse:syn2790911/wiki/235387).

### Neuroimaging data
Imaging data (PET, MRI, MEG) will be collected at each site and uploaded in DICOM format directly to the DPUK Imaging Informatics infrastructure (https://info.dpuk.org), which is based on the XNAT technology (https://www.xnat.org). DFP will be set up as a stand-alone 'node' on the central hub, which is physically located at the Farr Institute in Swansea. Image analysis pipelines, including quality control, will be run either through the XNAT infrastructure directly, or by downloading data for processing. Imaging-derived phenotypes will be extracted and uploaded to central data repository.

### Patient and public involvement
Patients and the public were involved in the study design through the pilot DFP study. Proceeding with the full study was contingent on the pilot demonstrating satisfactory participant acceptability of such an intensive study. Our results showed that there was a high degree of acceptability for all measures and that participants would have been willing to enter a similar study. In addition, we collected feedback on the study procedures through questionnaires, poststudy interviews as well as a focus group (the latter two conducted in collaboration with the Alzheimer's Society). This led to us change the study design to allow more flexibility for the role of study partners, shortening the PET scan duration and ensuring appointments are booked as much in advance as possible (see Section 'Deep and Frequent Phenotyping: a pilot study' for more details on this work). For the full study we will continue to partner with the Alzheimer's Society for the generation of information materials and videos explaining study procedures. We will also seek the Alzheimer's Society's lay panel input on monitoring study progress. Finally, we are planning to incorporate a qualitative research workstream aiming to understand participant engagement and burden as well as explore the participants' understanding and tolerability of processes related to biomarkers and precision medicine.

## SUMMARY
The DFP study is a major collaborative effort as part of Dementias Platform UK, bringing together both academic and industrial partners aimed at creating the most in-depth collection of fluid, clinical, imaging and digital biomarkers for prodromal AD. The feasibility and acceptability of such intense phenotyping was established through a pilot study that was deemed a success through predefined criteria. The primary outcomes of the study

will provide the means to identify target populations and assess outcomes in early phase, proof-of-concept clinical trials in AD more effectively than currently possible. Success in doing so would speed up trials, reduce costs, increase productivity and ultimately contribute to more effective pipelines for early clinical drug development. The programme will also generate a rich dataset that we intended to make available for use by the scientific community.

**Author affiliations**
[1]Department of Psychiatry, University of Oxford, Oxford, UK
[2]IMED Neuroscience, AstraZeneca UK Ltd, Cambridge, Cambridgeshire, UK
[3]Invicro, London, UK
[4]Department of Medicine, Imperial College London, London, UK
[5]Department of Psychiatry, University of Cambridge, Cambridge, Cambridgeshire, UK
[6]Department of Clinical Neurosciences, University of Cambridge, Cambridge, Cambridgeshire, UK
[7]Institute of Neuroscience, Newcastle University, Newcastle upon Tyne, UK
[8]Department of Clinical Neurosciences, University of Cambridge, Cambridge, UK
[9]MRC Cognition and Brain Sciences Unit, Cambridge, Cambridgeshire, UK
[10]MRC Biostatistics Unit, University of Cambridge, Cambridge, UK
[11]Department of Neurology, Imperial College London Faculty of Medicine, London, UK
[12]Medical School, University of Exeter, Exeter, UK
[13]Department of Psychiatry, Centre for Clinical Brain Sciences, University of Edinburgh, Edinburgh, UK
[14]Manchester Academic Health Sciences Centre, Institute of Brain, Behaviour, and Mental Health, Manchester, UK
[15]Queen's University Belfast, Belfast, UK
[16]Alzheimer's Society, London, London, UK
[17]Leonard Wolfson Experimental Neurology Centre, University College London Institute of Neurology, London, London, UK

**Acknowledgements**  The authors would like to thank members of the research teams and collaborators that participated in pilot Deep and Frequent Phenotyping study: Kia Nobre, Giovanna Zamboni, Clare O'Donoghue, Aimie Gornall, Andrew Quinn (University of Oxford); Basil Ridha, Martin Rossor, Nick Fox, Cathleen Chabo, Elizabeth Donnachie, Natalia Budnik, Lajos Csincsik (University College London); Tunde Peto (Moorfields Eye Hospital NHS Foundation Trust); Robert Howard, Clive Ballard, Dominic ffytche, Leeza Almedom, Alison Farrand (King's College London); Paul Matthews, Craig Ritchie, Genevieve Morrison, Tahira Arshad, Vanessa Raymont (Imperial College London); John O'Brien, Patricia Vasquez-Rodriguez, Alicia Wilcox (University of Cambridge); John-Paul Taylor, Kathryn Walker, Ginette Cass, Aodhan Hickey (Newcastle University); Doug Brown (Alzheimer's Society); Ilan Rabiner, Azadeh Firouzian (Invicro); Derek Hill, John Hall (Ixico). The authors would like to thank all participants and their families, the clinical research nurses at participating Clinical Research Facilities and Invicro, the PET technicians and radiochemists, the MRI radiographers for their cooperation and support to the pilot study. For the pilot study, the authors would like to thank Avid Radiopharmaceuticals for the provision of precursor for [$^{18}$F]AV1451 along with associated information on radio tracer production as well as Cambridge Cognition for the training and expertise in using the CANTAB machine. For the main Deep and Frequent Phenotyping study, please see supplementary files for a full list of all contributors to the design and local site set-up. For the main study, the authors would also like to thank the NIHR Oxford Health Biomedical Research Centre and the contribution of our industry partners Aridhia (data management), Exprodo (trial management system), Trialspark (website and participant recruitment software), Berry Consultants (statistical analysis), NeuroVision Imaging and OPTOS (ophthalmology equipment). The Mezurio smartphone application was developed by the University of Oxford Big Data Institute and supported through grants from Roche Holding AG and Eli Lilly. Sea Hero Quest was developed in a collaboration between Deutsche Telekom, University College London (UCL), The University of East Anglia (UEA) and Alzheimer's Research UK.

**Contributors**  SL is the Chief Investigator of the study and has overall responsibility for the study. IK, JL and TC (AstraZeneca, Industry Lead) led the study protocol generation. The following authors have led the individual modality sections of the protocol: CM (imaging), JBR (MEG), RG (PET), DLT (MRI), BS (cognition), LR (gait and wearable modality), IL (ophthalmology), BT (statistics); the following authors lead DFP sites: IK (University of Oxford), DC (University of Cambridge), CB (King's College London and University of Exeter), AJT (University of Newcastle), VR (University of Edinburgh), IL (University of Manchester), PM (Imperial College London). MM has contributed to the protocol as a representative of Alzheimer's Society. IC coordinated input into the protocol from industry partners. CWR is the Chief Investigator of the EPAD platform and the current study shares part of the EPAD protocol. JG is the director of Dementia Platforms UK and has contributed to the integration of the current study to the DPUK infrastructure.

**Funding**  The study is funded through a grant (MR/N029941/1) from the National Institute for Health Research (NIHR) and the Medical Research Council (MRC). Regular reports regarding the study will be made to the MRC Oversight Board. In-kind support for this study is provided from the following collaborators: Alzheimer's Society, Aridhia, Berry Associates, Cambridge Cognition, Exprodo, TrialSpark, AstraZeneca. NeuroVision Imaging are providing equipment and funding to run additional ophthalmology assessments. DLT acknowledges funding for work relevant to this study protocol through UCL Leonard Wolfson Experimental Neurology Centre (PR/ylr/18575) and an ARUK Network Acceleration award for cross-platform harmonisation of 3T MRI protocols for dementia (ARUK-NAS2016B-2). JBR is supported by the Wellcome Trust (103838).

**Competing interests**  AJT acknowledges grants from General Electric Healthcare, outside the submitted work. BS declares consultancy work for Cambridge Cognition and Mundopharma, outside the submitted work. CM declares that she is married to the management director of Exprodo whose systems are used in the data management of the study. CWR declares being on advisory boards for or receiving paid lectures from Pfizer, MSD, Actinogen, Roche, Nutricia, Allergan, Lundbeck, Biogen, Eisai, Prana, GSK, AstraZeneca, Eli Lilly, Janssen, GE, Piramal, Sanofi, Shire; CER also declares research funding from MSD, Eisai, Janssen, Actinogen, Eisai, Biogen, Takeda. CB reports grants and personal fees from Acadia pharmaceutical company, grants and personal fees from Lundbeck, personal fees from Roche, personal fees from Otusaka, personal fees from Novartis, personal fees from Eli Lilly, personal fees from Pfizer, outside the submitted work. IL declares an unrestricted grant from OPTOS. JG declares being the recipient of a Medical Research Council grant as the Chief Investigator of Dementias Platform UK, which is directly supporting the DFP study. JBR reports grants from Medical Research Council, grants from National Institute for Health Research, during the conduct of the study; grants from Wellcome Trust, grants from Dementias Platform UK, grants from AZ-Medimmune, personal fees from Asceneuron , non-financial support from AVID, grants from PSP Association, grants from McDonnell Foundation, outside the submitted work. PM declares a grant from Shire Pharmaceuticals, whereby he is provided with study medication free of charge. RG declares consultancy work for AbbVie, Biogen, Cerveau. SL reports activities outside the submitted work (consultancy and speaker fees for Merck and Eaisi, respectively; unrelated grant from AstraZeneca and grants from EFPIA companies arising from multiple IMI funding) as well as an unpaid membership to the SomaLogic medical advisory board. During the course of submission of this article he has become an employee of Janssen Pharmaceuticals.

**Patient consent for publication**  Not required.

**Provenance and peer review**  Not commissioned; externally peer reviewed.

**Data sharing statement**  The article details results from the pilot Deep and Frequent Phenotyping study that support the conduct of the main Deep and Frequent Phenotyping study. Unpublished data from the pilot study are in the process of being made freely available to researchers through the Dementias Platform UK Data Portal: https://www.dementiasplatform.uk/.

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
