## [Reviewer comments · BMJ Open]

ARTICLE DETAILS

TITLE (PROVISIONAL)	Deep and Frequent Phenotyping Study Protocol: An Observational Study in Prodromal Alzheimer's Disease
AUTHORS	Koychev, Ivan; Lawson, Jennifer; Chessell, Tharani; Mackay, Clare; Gunn, Roger; Sahakian, Barbara; Rowe, James; Thomas, Alan J.; Rochester, Lynn; Chan, Dennis; Tom, Brian; Malhotra, Paresh; Ballard, Clive; Chessell, Iain; Ritchie, Craig; Raymont, Vanessa; Leroi, Iracema; Lengyel, Imre; Murray, Matt; Thomas, David; Gallacher, John; Lovestone, Simon

VERSION 1 – REVIEW

REVIEWER	Michael Wagner DZNE, German Center for Neurodegenerative Diseases, Bonn
REVIEW RETURNED	03-Aug-2018

GENERAL COMMENTS	I congratulate the investigators for getting this timely study launched! The multiple and frequently applied behavioral and cognitive measures are a particularly strong feature of the study. Some details were missing or unclear in the protocol and should be clarified in a revision: 1. Abstract: The second sentence mentions clinical trials in preclinical disease – however the present study appears to be a prodromal phase study. The term preclinical is misleading here. In the second sentence, one could rephrase to „early disease stages“2. The selection of participants /definition of groups is not entirely clear. Several enrichment strategies are used: MCI patients from memory clinics, ApoE4 carriers, family history, poor CANTAB PAL. A) Which measures are to be used to quantify the degree of cognitive impairment at baseline? Will all subjects have amnesic MCI according to clinical standards? Or will the study population be a „mixed bag“? B) How are the controls recruited? If the controls come from the public advertisements, they seem to have at least a family history of AD, and perform below the 30th percentile of the CERAD PAL (maybe even below the 10th percentile?). It is unclear what this group can control for, as they may still be at increased risk for AD (by virtue of FH and ApoE and PAL) and just below conventional Amyloid cutoffs. If a continuum of subjects with differing degrees of biomarker abnormalities/risk is to be studied, then the term controls should be avoided. How about speaking of Amyloid positive and Amyloid negative participants? This seems to be the critical distinction between both groups.3. The outcomes should be clearly defined. As you have so many cognitive outcomes, which one is critical for the definition of „those
---

	who had a worsening in cognitive functioning“ (p. 28)? What exactly constitutes the disease change which you want to track? Only few participants will have a change in disease state according to conventional criteria (e.g. CDR global ≥ 1) within 12 months. 4. Many studies in the prodromal phase have shown that functional changes (reported by informants) are capturing important variance of disease progression, and cognitive-functional composites have been proposed as most sensitive outcome measures for clinical trials (e.g. ADCOMS, Wang et al., J Neurol Neurosurg Psychiatry. 2016 Sep;87(9):993-9.). Is the presence of a study partner/informant an inclusion criterion? It is not listed with the criteria in the Supplement. If not, the CDR cannot be done, and you cannot benchmark your results against the ADCOMS or against the CDR-Sum of Boxes, which are the champions to beat in a study aiming to „identify biomarkers of greater efficiency“. If yes, then these measures should be computed as a reference. 5. It seems that between-group analyses (declining vs. stable subjects) are the initial statistical analysis. However, whatever outcome you choose for defining the two groups, defining groups by applying a study-specific cutoff to a continuum of decline (and measurement error) seems to me somewhat arbitrary; if this approach has been successfully applied before, references should be given. An alternative might be mixed model analyses of the whole group (or of the Amyloid positive group only, as this group likely reflects the target group for clinical trials), examining internal responsiveness (mean to standard deviation ratio for each change measure; this will give a ranking of sensitivity to change) and external responsiveness (by correlations with e.g. the ADCOMS or the CDR-SOB, this will help to prioritize those novel measures most strongly associated with published measures of disease progression). Minor issues: 6. Deleting the PAL screening information (p. 16) for those entering the study does not seem to make sense in such a study – is this a data protection issue ? 7. Give the ADNI references used for the power calculation.
--	--

REVIEWER	Tharick A. Pascoal McGill University, Canada
REVIEW RETURNED	20-Aug-2018

GENERAL COMMENTS	The authors present a clear and well-written manuscript describing an important initiative in which they will access prodromal-AD and elderly controls (both groups enriched with and without presence AD pathophysiology, respectively) with multiple biomarkers in short periods of time to identify markers capable of tracking AD progression in early stages. The study is well designed and conducted by a strong and well-known group of researchers who contemplate various areas of knowledge necessary to conduct a study like this. This report is of enormous interest to the AD community and I do not have major concerns regarding the description of the study protocol, only minor comments. Minor comments:
---

	1 – Since one of the strengths of the present study is to collect biomarkers repeatedly in order to have a more representative and reliable disease progression data, the author could elaborate a bit more on how they will address one of the main issues related to multicenter designs that are dealing with the harmonization of the data across sites: cognition, different scanners across sites, CSF (specific time for LPs, since the authors can decide to assess biomarkers with circadian variation), etc. 2- Dynamic PET acquisitions are sometimes a burden and acquiring static PET data may help to reduce participant attrition. However, if the author's objective is to track disease progression with more representative data, could the authors elaborate on the reasons they decided against acquiring PET scans dynamically? Understanding this decision rationale would be helpful to other researchers facing the same questions in their study designs. 3- The authors mention that the acquisition of PET will be "over 30 minutes". Perhaps, they should just mention that it would be according to the tracer. It seems that most of the NFT ligands would require as little as 20 min acquisition. 3- It is not clear in the manuscript if the data sharing will occur 1 year after the last participant performs the last evaluation or 1 year after the researchers complete their analyses. If the correct answer is the latter, more details should be provided regarding the timeline of this process. 4- Could the author elaborate on how the thresholds for biomarker positivity will be defined without AD patients in the study? CSF: Will the CSF be quantified in a center that has already defined this previously? PET: "Aβ positivity will be determined based on the value of the SUVR of the cortex". Will they use an image pipeline with a previously published cut-off? Will they first visually define Aβ positively? Will the authors stratify patients on tau positive and negative? If so, how?
--	---

VERSION 1 – AUTHOR RESPONSE

Reviewer Name: Michael Wagner

I congratulate the investigators for getting this timely study launched! The multiple and frequently applied behavioral and cognitive measures are a particularly strong feature of the study. Some details were missing or unclear in the protocol and should be clarified in a revision:

1. Abstract: The second sentence mentions clinical trials in preclinical disease – however the present study appears to be a prodromal phase study. The term preclinical is misleading here. In the second sentence, one could rephrase to „early disease stages“

Author comments: Thank you for pointing out this inconsistency. We have amended the text along your suggestion.

2. The selection of participants /definition of groups is not entirely clear. Several enrichment strategies are used: MCI patients from memory clinics, ApoE4 carriers, family history, poor CANTAB PAL. A) Which measures are to be used to quantify the degree of cognitive impairment at baseline? Will all

subjects have amnesic MCI according to clinical standards? Or will the study population be a „mixed bag“? B) How are the controls recruited? If the controls come from the public advertisements, they seem to have at least a family history of AD, and perform below the 30th percentile of the CERAD PAL (maybe even below the 10th percentile?). It is unclear what this group can control for, as they may still be at increased risk for AD (by virtue of FH and ApoE and PAL) and just below conventional Amyloid cutoffs. If a continuum of subjects with differing degrees of biomarker abnormalities/risk is to be studied, then the term controls should be avoided. How about speaking of Amyloid positive and Amyloid negative participants? This seems to be the critical distinction between both groups. Author comments: We are sorry that this was not clear and welcome the opportunity to make it so. We have now made the distinction between the enrichment strategy and the inclusion criteria more obvious. We have clarified the processes of the different enrichment strategies which are only different in as much as the sources of potential participants have differing pre-existing data availability.

The only inclusion criterion for the study itself is based on amyloid determined by either PET or CSF and we hope this is now more obvious. The other assessments, of family history, cognitive impairment and APOE4 status are enrichment strategies to increase the efficiency of the recruitment process and reduce unnecessary intensive/invasive tests of amyloid.

We agree that it is more appropriate to speak of amyloid positive vs amyloid negative participants as opposed to controls and have adjusted the text accordingly

In response to the reviewer's specific questions: A) Where participants are recruited from existing cohorts, existing cognitive scores will be assessed and enrichment will occur at a 4:1 ratio based on the mean scores for that cohort. All other participants (those recruited through public advertisements or memory clinics) will complete the PAL online test at pre-screening. Abnormal scores will be determined based on PAL's normative database (range between 1 and 1.5 SD below the mean) and again will be invited on a 4:1 ratio basis. In both cohort- and public advertisement-based recruitment subclinical will be defined as being in the range of 1-1.5 SD below the mean). B) Amyloid negative individuals will enter the study through the same routes as amyloid positive individuals. We agree with Professor Wagner's point that although we expect amyloid negative individuals to be on a different trajectory relative to amyloid positive ones we should avoid the term 'controls'. We have amended the text to reflect the above.

3. The outcomes should be clearly defined. As you have so many cognitive outcomes, which one is critical for the definition of „those who had a worsening in cognitive functioning“ (p. 28)? What exactly constitutes the disease change which you want to track? Only few participants will have a change in disease state according to conventional criteria (e.g. CDR global ≥ 1) within 12 months.

Author comments: The reviewer raises an important point – we agree it is unlikely that we will see significant change in classical dementia trial outcome measures, although we have enriched for participants with APOE4. We will be considering the 1-year change in the CANTAB cognitive co-primary outcome measures (Adjusted Errors on the Paired Associates Learning Task and Between Search Errors on the Spatial Working Memory Task) where clinically meaningful worsening will be defined as a reduction by 3 and 1 points respectively. We have amended the text to reflect this.

4. Many studies in the prodromal phase have shown that functional changes (reported by informants) are capturing important variance of disease progression, and cognitive-functional composites have been proposed as most sensitive outcome measures for clinical trials (e.g. ADCOMS, Wang et al., J Neurol Neurosurg Psychiatry. 2016 Sep;87(9):993-9.). Is the presence of a study partner/informant an inclusion criterion? It is not listed with the criteria in the Supplement. If not, the CDR cannot be done, and you cannot benchmark your results against the ADCOMS or against the CDR-Sum of Boxes,

which are the champions to beat in a study aiming to „identify biomarkers of greater efficiency“. If yes, then these measures should be computed as a reference.

Author comments: Based on feedback from participants the pilot study we decided not to make availability of study partners an inclusion criteria. We will however be seeking to complete the CDR with informers either in person or by phone. We anticipate that this would not be possible only for a minority of participants.

5. It seems that between-group analyses (declining vs. stable subjects) are the initial statistical analysis. However, whatever outcome you choose for defining the two groups, defining groups by applying a study-specific cutoff to a continuum of decline (and measurement error) seems to me somewhat arbitrary; if this approach has been successfully applied before, references should be given. An alternative might be mixed model analyses of the whole group (or of the Amyloid positive group only, as this group likely reflects the target group for clinical trials), examining internal responsiveness (mean to standard deviation ratio for each change measure; this will give a ranking of sensitivity to change) and external responsiveness (by correlations with e.g. the ADCOMS or the CDR-SOB, this will help to prioritize those novel measures most strongly associated with published measures of disease progression).

Author comments: We agree with the reviewer that loss of efficiency/power will result from dichotomising into two groups (stable and declining). Our first wave analysis is based on this dichotomy primarily because our power calculation was based on the defining of these two groups. We intend to in our further investigations to treat the cognitive measure (e.g. CANTAB composite memory score) as a longitudinal continuous outcome and model it using regression methodology, which includes mixed modelling methodology. We have now made this explicit in the text.

Minor issues:

6. Deleting the PAL screening information (p. 16) for those entering the study does not seem to make sense in such a study – is this a data protection issue ?

Author comments: This is indeed a data protection issue – we have been advised that keeping the data on the Cambridge Cognition servers may be problematic.

7. Give the ADNI references (used for the power calculation).

Author comments: The ADNI reference (McEvoy LK, Edland SD, Holland D, Hagler JD, Roddey JC, Fennema-Notestine C, Salmon DP, Koyama AK, Aisen PS, Brewer JB, Dale AM. Neuroimaging enrichment strategy for secondary prevention trials in Alzheimer disease. Alzheimer disease and associated disorders. 2010;24(3):269-77) is now included.

Reviewer: 2

Reviewer Name: Tharick A. Pascoal

Institution and Country: McGill University, Canada

Please state any competing interests or state 'None declared': None declared

Please leave your comments for the authors below

The authors present a clear and well-written manuscript describing an important initiative in which they will assess prodromal-AD and elderly controls (both groups enriched with and without presence AD pathophysiology, respectively) with multiple biomarkers in short periods of time to identify markers capable of tracking AD progression in early stages. The study is well designed and conducted by a strong and well-known group of researchers who contemplate various areas of knowledge necessary to conduct a study like this. This report is of enormous interest to the AD community and I do not have major concerns regarding the description of the study protocol, only minor comments.

Minor comments:

1 – Since one of the strengths of the present study is to collect biomarkers repeatedly in order to have a more representative and reliable disease progression data, the author could elaborate a bit more on how they will address one of the main issues related to multicenter designs that are dealing with the harmonization of the data across sites: cognition, different scanners across sites, CSF (specific time for LPs, since the authors can decide to assess biomarkers with circadian variation), etc.

Author comments: The reviewer is correct in pointing out that harmonization, particularly in regards to CSF and neuroimaging, is a major challenge for a study of this type. This was one of the outcomes of the pilot DFP study which demonstrated no significant between site differences for each of the study modalities. CSF will be measured in a single batch at a central facility (Prof Henrik Zetterberg's group). Due to the complex nature of the study logistics it has not been possible to set a specific time for the LP but the time will be recorded for post-hoc analysis of effects. The MRI protocol has been modified for the network of PET-MR scanners established by Dementias Platform UK. Prior work on this network across sites with different scanner manufacturers produced an optimised protocol, and this will be augmented by a 'traveling heads' study to establish concordance between sites. Finally, the teams responsible for the potentially unfamiliar CANTAB battery and gait assessments will be providing training to the study research assistants as done in the pilot study.

2- Dynamic PET acquisitions are sometimes a burden and acquiring static PET data may help to reduce participant attrition. However, if the author's objective is to track disease progression with more representative data, could the authors elaborate on the reasons they decided against acquiring PET scans dynamically? Understanding this decision rationale would be helpful to other researchers facing the same questions in their study designs.

Author comments: We collected dynamic PET alongside the static acquisition in the pilot DFP study to enable the investigation of more simplified static acquisitions that require much shorter scanning times for the subjects. Whilst dynamic PET scans can be desirable, for studies such as this one involving more challenging patient populations we decided that dynamic scans would not be optimal in such a logistically challenging study due to their increased duration (associated with increased burden to patients, difficulties in arranging time slots and increasing scan costs).

3- The authors mention that the acquisition of PET will be "over 30 minutes". Perhaps, they should just mention that it would be according to the tracer. It seems that most of the NFT ligands would require as little as 20 min acquisition.

Author comments: We have decided to acquire 30 mins of static data which balances increased imaging statistics whilst limiting the time a subject spends in the scanner.

3- It is not clear in the manuscript if the data sharing will occur 1 year after the last participant performs the last evaluation or 1 year after the researchers complete their analyses. If the correct answer is the latter, more details should be provided regarding the timeline of this process.

Author comments: Access to the data will be facilitated a year after data has been prepared for analysis. We anticipate this preparatory stage to last no more than one year. Team-exclusive access to the data will be for one year after that.

4- Could the author elaborate on how the thresholds for biomarker positivity will be defined without AD patients in the study? CSF: Will the CSF be quantified in a center that has already defined this previously? PET: "A β positivity will be determined based on the value of the SUVR of the cortex". Will they use an image pipeline with a previously published cut-off? Will they first visually define A β positively? Will the authors stratify patients on tau positive and negative? If so, how?

Author comments: The CSF will be analysed at Henrik Zetterberg's group who have defined well-validated cut-offs for amyloid for their assays. The A β +/- will be determined automatically by calculating the cortical SUVR and applying a tracer specific threshold which will be selected based on

literature values. This stratification will be used for screening purposes only. We do not plan to stratify patients into tau positive and negative in the first instance but instead intend to use tau values as continuous variables. Secondary analyses may use tau stratification depending on the availability of validated cut-offs for the relevant CSF assays and tau ligands.

VERSION 2 – REVIEW

REVIEWER	Michael Wagner DZNE, German Center for Neurodegenerative Diseases, Bonn, Germany
REVIEW RETURNED	04-Oct-2018

GENERAL COMMENTS	The authors have very well responded to all issues raised, two minor issues and one typo remain:  1. It has now been made clear that all participants (in the second step of participant selection) will undergo the online CANTAB PAL. Sub-clinical cognitive impairment is now clearly defined as 1 SD - 1.5 SD PAL deficit (there is a typo: "between 1 and 1 SD" in the last line of page 13, which needs correction). However, "no apparent impairment" is not defined in this paragraph (page 14, first line). It appears that subjects between 0.5 SD deficit (roughly 30th percentile) and 1 SD deficit (roughly 15th percentile) are considered to have "no apparent impairment", as the 30th percentile is referred to in the cognition section on page 15. Or are only high PAL scorers (the best 30 percent) considered as "no apparent impairment"? Please clarify, and consider using consistently only percentiles or SD deficits in defining the putatively memory-impaired and non-impaired subjects to be invited for the screening session in a 4:1 ratio. 2. Another point, introduced in the revision in response to my question, should be further clarified: On page 27 of the revised manuscript, the authors now write: "Cognitive functioning will be defined through the cognitive co-primary outcome measures of i) Adjusted Errors on the Paired Associates Learning Task and ii) Between Search Errors on the Spatial Working Memory Task. Clinically meaningful deterioration will be defined as a loss of 3 and 1 points respectively over the 12-month study duration." Please specify whether deterioration in EITHER measure or in BOTH measures will be used for group definition, and give a reason for the clinical meaningfulness of the cutoffs (e.g. by referring to changes observed in previous longitudinal MCI studies with these CANTAB tasks, or by referring to a reliable change score based on retest-reliability).
--

VERSION 2 – AUTHOR RESPONSE

Reviewer Name: Michael Wagner

The authors have very well responded to all issues raised, two minor issues and one typo remain:

1. It has now been made clear that all participants (in the second step of participant selection) will undergo the online CANTAB PAL. Sub-clinical cognitive impairment is now clearly defined as 1 SD - 1.5 SD PAL deficit (there is a typo: "between 1 and 1 SD" in the last line of page 13, which needs correction). However, "no apparent impairment" is not defined in this paragraph (page 14, first line). It appears that subjects between 0.5 SD deficit (roughly 30th percentile) and 1 SD deficit (roughly 15th percentile) are considered to have "no apparent impairment", as the 30th percentile is referred to in the cognition section on page 15. Or are only high PAL scorers (the best 30 percent) considered as "no apparent impairment"? Please clarify, and consider using consistently only percentiles or SD deficits in defining the putatively memory-impaired and non-impaired subjects to be invited for the screening session in a 4:1 ratio.

Authors' comments: We agree that for clarity it is better to use either SDs and percentiles and now express the range in SDs only. We define no apparent cognitive impairment as PAL scores ranging between 0.5 and 1 SD. We have now corrected the text accordingly on pages 13 and 14. Also many thanks for pointing out the type on page 13 – this has been corrected.

2. Another point, introduced in the revision in response to my question, should be further clarified: On page 27 of the revised manuscript, the authors now write: "Cognitive functioning will be defined through the cognitive co-primary outcome measures of i) Adjusted Errors on the Paired Associates Learning Task and ii) Between Search Errors on the Spatial Working Memory Task. Clinically meaningful deterioration will be defined as a loss of 3 and 1 points respectively over the 12-month study duration."

Please specify whether deterioration in EITHER measure or in BOTH measures will be used for group definition, and give a reason for the clinical meaningfulness of the cutoffs (e.g. by referring to changes observed in previous longitudinal MCI studies with these CANTAB tasks, or by referring to a reliable change score based on retest-reliability).

Authors' comments: Thank for pointing out this inconsistency. Clinically meaningful deterioration is defined as a decrease in either score. The rationale for choosing these cut-offs is based on a study showing that in amyloid-positive and APOE E4 carriers an average 3-point change in PAL and 1 point in SWM has been observed for a sample of MCI patients who transition to Alzheimer's disease (Nathan 2016). We have amended the manuscript accordingly.